# Infection prevention and control measures to reduce the transmission of mpox: A systematic review

**Rebecca Kuehn**[1]*, **Tilly Fox**[1], **Gordon Guyatt**[2], **Vittoria Lutje**[1], **Susan Gould**[1]

**1** Department of Clinical Sciences, Liverpool School of Tropical Medicine, Liverpool, United Kingdom,
**2** Department of Health Research Methods Evidence and Impact, McMaster University, Hamilton, Canada

\* Rebecca.kuehn@lstmed.ac.uk

**Data Availability Statement:** All relevant data for this study are available within the paper and its Supporting Information files.

## Abstract

### Objectives

To make inferences regarding the effectiveness of respiratory interventions and case isolation measures in reducing or preventing the transmission of mpox based on synthesis of available literature.

### Methods

The WHO Clinical Management and Infection Prevention and Control 2022 guideline and droplet precautions in healthcare facilities and home isolation infection prevention control measures for patients with mpox. We conducted a systematic review that included a broad search of five electronic databases. In a two-stage process, we initially sought only randomized controlled trials and observational comparative studies; when the search failed to yield eligible studies, the subsequent search included all study designs including clinical and environmental sampling studies.

### Results

No studies were identified that directly addressed airborne and droplet precautions and home isolation infection prevention control measures. To inform the review questions the review team synthesized route of transmission data in mpox. There were 2366/4309 (54.9%) cases in which investigators identified mpox infection occurring following transmission through direct physical sexual contact. There were no reported mpox cases in which investigators identified inhalation as a single route of transmission. There were 2/4309 cases in which investigators identified fomite as a single route of transmission. Clinical and environmental sampling studies isolated mpox virus in a minority of saliva, oropharangeal swabs, mpox skin lesions, and hospital room air.

### Conclusions

Current findings provide compelling evidence that transmission of mpox occurs through direct physical contact. Because investigators have not reported any cases of transmission

**Funding:** RK, TF and VL are supported by the Research, Evidence and Development Initiative (READ-It). The editorial base of the Cochrane Infectious Diseases Group and READ-It (project number 300342-104) are funded by UK aid from the UK government for the benefit of low- and middle-income countries (project number 300342-104). The views expressed do not necessarily reflect the UK government's official policies. SG is supported by the National Institute for Health Research Health Protection Research Unit in Emerging and Zoonotic Infections. SG is also supported by the Research, Evidence, and Development Initiative, which is funded by UK aid from the UK government (project number 300342–104). The views expressed here do not necessarily reflect the UK government's official policies.

**Competing interests:** The authors have declared that no competing interests exist.

via inhalation alone, the impact of airborne and droplet infection prevention control measures in reducing transmission will be minimal. Avoiding physical contact with others, covering mpox lesions and wearing a medical mask is likely to reduce onward mpox transmission; there may be minimal reduction in transmission from additionally physically isolating patients with mild disease at home.

## Introduction

Mpox is a zoonotic disease caused by mpox virus, an enveloped double-stranded DNA virus in the *Orthopoxvirus* genus of the *Poxviridae* family. The World Health Organization (WHO) declared mpox (then termed monkeypox) a Public Health Emergency of International Concern (PHEIC) between July 2022 and May 2023. The 2022 mpox outbreak was associated with sustained human-to-human transmission that had not been previously described [1]. Historically, mpox occurred primarily in Central and West Africa, with infection commonly reported in persons who had contact with probable animal reservoirs, cases of secondary transmission were most often reported in household contacts [2]. Increased incidence in endemic areas of central and West Africa over the past forty to fifty years coincides with the cessation of smallpox vaccination and eradication programmes [3]. There are two distinct clades of mpox virus, Clade I (formerly known as the Central African or Congo Basin clade) and Clade II (formerly known as the West African clade). Clade II consists of two subclades, Clade IIa and Clade IIb.

Mpox incubates between five to 21 days and typically presents symptoms in two stages: the invasion period lasting from zero to five days characterized by fever, headache, lymphadenopathy, back pain, myalgia, and asthenia; following this, skin symptoms may appear between one to three days from onset of fever, with a rash evolving from macules to papules, vesicles, pustules and then crusts, often affecting the face, extremities, oral mucous membranes, and genitalia [4].

Suspected or confirmed transmission routes of mpox include direct physical contact with an infected patient (non-sexual physical contact or sexual physical contact), indirect contact (fomite transmission), inhalation of fomites or infectious droplets, inoculation and transplacental transmission [2]. Human infection is also possible from contact with infected animals (scratches, bites, preparing, eating or using infected meat and animal products) [2]. Areas of uncertainty exist concerning the potential for asymptomatic transmission or the transmission potential of other possible routes, such as breastmilk, semen, vaginal fluids, urine, faeces or insect vectors.

There is a need for interventions to prevent the transmission of mpox. The effectiveness of any IPC measures for mpox depends on route(s) of transmission of mpox virus. The WHO Clinical Management and Infection Prevention and Control 2022 guideline development group developed two research questions concerning airborne and droplet IPC interventions and one question concerning case physical isolation interventions in mpox. It was expected that scarce evidence, if any, from randomized controlled trials or comparative interventional trials would exist to inform the research questions. As such, it was anticipated that the review questions could be informed indirectly using data on the number of incident cases of mpox by route of transmission and clinical and environmental sampling studies demonstrating viral culture positivity. This is based on the inference that mpox infections will be reduced through interventions targeting the most frequently reported routes of transmission of mpox virus. In the case of IPC interventions targeting airborne and droplet precautions and interventions

targeting physical home isolation of cases, if there are a significant number of cases of infection transmitted by inhalation of virus aerosols, droplets or fomites and/or strong viral culture positivity from samples representative of these routes of transmission, interventions directed at preventing transmission via these route(s) would be of importance. Therefore, we conducted a systematic review in two stages—the first stage seeks evidence from comparative interventional trials, and the second stage seeks evidence from all other study designs—to inform the infection prevention and control guideline recommendations regarding use of airborne and droplet precautions to mpox transmission and physical isolation of mpox patients.

## Objective

- To make inferences regarding the effectiveness of airborne and droplet interventions and case home isolation measures in reducing or preventing the transmission of mpox based on synthesis of available literature.

## Review questions

The review followed two pre-planned protocols (available on request to corresponding author). The review addresses three research questions developed by the WHO Clinical Management and Infection Prevention and Control 2022 guideline development group [5] (for full review questions see S1–S3 Tables):

1. Does healthcare worker use of respirator versus a medical mask when interacting with a confirmed/suspect mpox patient during the infectious period reduce mpox infections?

2. Does the use of an airborne precaution room versus an adequately ventilated room in a healthcare facility for a mpox patient in the infectious period reduce mpox infection in health workers or patients?

3. Does isolating a person with mpox until all lesions are fully healed versus not isolating reduce mpox infections?

## Review stage one: Review of infection prevention and control interventions for preventing mpox infection

In the first stage, we aimed to synthesize evidence concerning the review question interventions from available interventional comparative studies.

## Methods

**Inclusion criteria.**   Types of studies

- RCTs, controlled before-and-after studies, observational comparative studies in participants with confirmed mpox or exposed to mpox virus.

- RCTs or observational comparative studies in participants with exposure to or confirmed mpox-like virus infection.

  Population
  Humans with laboratory confirmed mpox infection; or
  humans with laboratory confirmed mpox-like infection; or
  humans with symptoms consistent with mpox and exposure to a laboratory confirmed mpox infection or mpox-like infection.

Mpox-like infection is defined for the purposes of this review as infection due to orthopox viruses other than mpox that are capable of human-to-human transmission, namely buffalo-pox, cowpox, vaccinia, and variola.

Types of interventions

1. Airborne and droplet precautions including medical masks or use of respirators.

2. Personal contact precautions including use of gloves, gowns, eye protection.

3. Isolation of cases

4. Ventilation including natural, mechanical, negative pressure gradient, positive pressure ventilated lobby.

The administration of any type of vaccine to health care workers or contacts of mpox or mpox-like virus confirmed patients was not considered an intervention type.

Control

- No intervention or;

- Any different intervention measure used as a comparator to the intervention group in the study.

Outcomes

1. Confirmed secondary mpox or mpox-like virus infection expressed as an absolute number or rate of secondary transmission.

2. All reported adverse effects related to the interventions.

Settings

All countries and the following contexts were eligible for this review: households, congregate-living, community and healthcare settings.

## Exclusion criteria

Studies were excluded if any of the following criteria were met:

1. Studies published in a language other than English.

2. Studies of designs other than RCTs, controlled before-and-after studies, or observational comparative studies for participants with confirmed mpox infection or exposure to mpox.

3. Studies of designs other than RCTs or observational comparative studies in participants with viruses other than viruses defined as mpox-like viruses.

4. Studies that did not include a review question-specific intervention to reduce or prevent the transmission of mpox or mpox-like viruses.

5. Studies conducted in animals.

## Literature search strategy

Using broad search terms including terms for mpox-like viruses and without date or language limits, the search in September 2022 included the following databases: MEDLINE (OVID), Embase (OVID), Biosis previews (Web of Science), CAB Abstracts (Web of Science), and Global Index Medicus (S1 Appendix). The review author team instituted a call to topic experts for papers concerning the review questions for relevant studies up to 15th December 2022.

### Selection of studies

The results of the literature searches were uploaded into Distiller SR software [6]. Screening of results was undertaken according to the Cochrane Collaboration's Rapid Review Methods due to the time-sensitivity of the review findings to inform guideline development [7]. Title and abstract screening of all studies identified in the literature searches was undertaken independently by multiple review authors; one author was required to assess a study as eligible for full text screening; two authors were required to assess a study as requiring exclusion. Full-text screening against the inclusion criteria was undertaken independently by multiple review authors. One author was required to assess a study as eligible for inclusion to data extraction; two review authors were required to assess a study for exclusion. Authors resolved disagreement at any stage by discussion.

### Data extraction and management

We planned for two authors to extract data from all included studies using a pre-piloted data extraction form within Distiller SR, however no eligible studies were identified.

### Risk of bias assessment

It was planned for two authors to independently conduct risk of bias assessments using the Cochrane Risk of bias 2 tool [8,9] for included randomized controlled trials and the ROBINS-I tool [10] for included non-randomised comparative studies; however no comparative trials were identified.

### Results

We did not identify any studies meeting the inclusion criteria for this stage of the review, so we moved to the second stage.

## Review stage two: Transmission route of mpox virus

The second stage of the review aimed to synthesize evidence on mpox infection as a result of transmission using a wider range of study designs that could indirectly inform the review questions by answering the following:

1. What is the proportion of incident cases of mpox disaggregated by route of transmission?

2. An incident case is defined as an individual changing from a state of non-disease to disease over a specific period of time, as reported by study authors.

3. What is the infectious period of mpox, disaggregated by route of transmission?

   The infectious period is defined as the number of days since the onset of symptoms.

### Methods

**Inclusion criteria.** <u>Population:</u> Human participant of any age with laboratory-confirmed mpox infection or symptoms consistent with mpox and exposure to a laboratory-confirmed mpox patient or exposure to a suspected human mpox case.

The WHO definition of a suspected case of mpox infection was used [11].

Laboratory-confirmed infection was defined as reverse-transcriptase polymerase chain reaction (RT-PCR) positive or viral culture positive.

Outcomes:

1. Mpox infection

   Type of study: any scientific article of any design including clinical and environmental sampling studies.
   Setting: All countries and all contexts.

## Exclusion criteria

Studies were excluded if any of the following criteria were met:

1. Studies not including a human case of laboratory confirmed mpox infection or exposure to a laboratory-confirmed mpox patient or exposure to a suspected mpox case.

2. Studies solely concerning animal-to-animal mpox transmission or animal-to-human transmission.

3. Studies not published in English.

4. Experimental laboratory transmission studies.

5. Studies meeting the inclusion criteria but not otherwise containing information relevant to the review questions.

## Literature search strategy and selection of studies

The review team used the same search strategy and methods for the selection of studies as for the first review stage (S1 Appendix).

## Data extraction and management

Two authors extracted data from all included studies using a pre-piloted data extraction form within Distiller SR. One author extracted all relevant data and the second author cross-checked all extracted data. Data was extracted concerning characteristics of the study participants including number of primary and/or secondary cases, country, year of study, setting of transmission (such as household, healthcare), clade of mpox, reported nature of contact of participants to a potential or confirmed course of mpox, study author reported modes or potential modes of transmission, and data concerning clinical or environmental sampling including sample type, and day of sample PCR or viral culture positivity from symptom onset.

## Appraisal of study quality

Risk of bias assessments are related to study design. We did not identify any applicable pre-existing tool to assess the risk of bias in included case reports and case series. We therefore constructed and piloted a series of questions to appraise the quality of included case reports and case series covering aspects of representativeness and comprehensiveness of included participants (S2 Appendix). Quality appraisal assessments were then undertaken independently by the review authors for all included studies; differences were resolved by discussion.

## Data synthesis

1. What is the proportion of incident cases of mpox disaggregated by route of transmission?
   Two authors independently categorized all reported human mpox cases from identified studies by route(s) of transmission. Authors resolved disagreement at any stage by discussion. Authors assigned the following route(s) of transmission to each case as applicable: direct sexual physical contact, direct non-sexual physical contact, fomite, inhalation, transplacental, needle-stick, ingestion or unknown.

The routes of transmission are defined as:

- Direct sexual physical contact: transmission occurring in the context of any type of sexual activity, including oral sex, penetrative anal or vaginal sex (insertive and receptive), or hand-to-genital contact.

- Direct non-sexual physical contact: direct physical touch with the exclusion of any sexual physical contact as defined above.

- Fomite: indirect contact transmission involving contact of a susceptible host with a contaminated object or surface.

- Inhalation: occurs when infectious particles, of any size (aerosols or droplets), travel through the air, enter and are deposited at any point within the respiratory tract of a (susceptible) person. This form of transmission can occur when the infectious particles have travelled either a short- or long-range from the infected person.

- Transplacental: transmission via the placenta from mother to foetus.

- Percutaneous injury: transmission via percutaneous injury with a contaminated object, such as a needle.

The review team then categorized all cases into one of three categories: (i) single route of transmission resulting in infection reasonably identified, (ii) multiple routes of transmission resulting in infection possible, and (iii) unknown route of transmission resulting in infection. The category of a single route of transmission applied when sufficient data concerning the case history, epidemiology, and/or clinical details was reported to reasonably judge that a single route of transmission leading to an mpox infection had occurred. The category of multiple possible routes of transmission applied when more than one route of transmission was judged as reasonably possible to result in an mpox infection based on reported information. Authors applied the category of unknown when there was insufficient information reported in the study to assign or hypothesize any route of transmission in a case of mpox infection.

The number of incident mpox cases for each route and category of transmission is reported as a whole number and percentage of the total. Data are presented for each route of transmission category, by mpox clade, and by route of transmission in the healthcare and household settings. Data that could inform the subgroups of the full review questions (S1–S3 Tables) within the research questions is summarized.

2. What is the infectious period of mpox, disaggregated by route of transmission?

Authors separated data from included studies into either human mpox clinical samples or environmental samples from an environment occupied by an mpox case. The review team assigned data concerning environmental air sampling, mask sampling, and upper respiratory tract clinical sampling to the review questions concerning prevention of airborne transmission. Data concerning environmental surface sampling and clinical sampling of active skin lesions were assigned to the review question concerning case isolation measures.

Within these categories the review team identified and summarized longitudinal studies and cross-sectional studies that attempted viral isolation. The number of samples for each category is presented.

## Summary of findings and assessment of certainty of the evidence

A summary of findings table is presented for each review question. Data to inform the outcome of mpox infection is inferred from the number of reported mpox cases by route of

transmission. Data from clinical and environmental sampling studies, representing lower quality evidence, is narratively summarized in the results section.

The rating of the certainty of evidence is based on the GRADE approach for observational studies following the GRADE guidance as recommended in the GRADE Handbook [12].

## Search results

The searches identified 2514 unique records. Authors assessed the full text of 725 studies; 122 studies were included and 603 studies were excluded. The study selection process is presented in S1 Fig.

## Included studies

114 studies reported cases of human-to-human mpox as a result of transmission [13–126]. 39 studies were conducted prior to the 2022 outbreak [15,16,19,21,24,31,32,35,38,43,46,49,50,53–56,62,66,69,73,74,76,77,80–83,95,99–102,116,117,120,123–125] and 75 studies were published during 2022 [13,14,17,18,20,22,23,25–30,33,34,36,37,39,40–42,44,45,47,48,51,52,57–61,63–65,67,68,70–72,75,78,79,84,85–94,96–98,103–115,118,119,121,122,126].

The region of acquisition of infection was reported as follows: 37 studies from Africa [15,19,21, 24,31,32,35,38,43,49,50,53–56,62,66,69,73,74,76,77,80–83,90,95,99–102,116,120,123–125], 1 study from the Eastern Mediterranean Region [121], 48 studies from Europe [13,14,18,20,22,23,25–29,33,36,37,39,40,42,44,45,47,48,51,52,57,58,60,61,67,72,78,79,84,86,88,89,92,94,97,98,104,109,111–115,122,126], 8 studies from North America [17,46,64,70,85,93,106,107], and 3 studies from South America [30,71,103]. 11 studies reported on cases acquired from multiple world regions [16,34,41,59,63,65,87,91,110,117,119], and 6 studies did not report the country of infection acquisition [68,75,96,105,108,118].

No studies concerning mpox-like viruses met the inclusion criteria.

There were 14 studies that provided data concerning clinical and environmental sampling [24,61,62,72,73,78,127–134]. Studies in which a denominator was not reported (that is, how many samples were taken in total) were not included.

## Quality assessment of included studies

The quality of assessment results are available in S3 Appendix. Studies generally differed in active case finding and in the reporting of sufficient information to hypothesize route(s) of transmission of mpox virus.

## Included cases

There were 4309 cases of human-to-human transmission resulting in mpox infection [113–126]. All cases of human-to-human acquisition were included; it was not always possible to determine whether a case was a primary or index case.

Table 1 displays included cases by age and gender. Most cases were males over the age of 18 years.It was not possible to disaggregate age and gender for 1062/4309 (24.6%) of cases (Table 1).

## Results: Review question 1 and 2

Review questions 1 and 2 are considered together as they concern airborne and droplet IPC interventions. The summary of findings for review question 1 are presented in Table 2 and the summary of findings for review question 2 are presented in Table 3.

**Table 1. Included cases by age and gender.**

| Gender | Under 18 years Number of cases / total cases | 18 years or older Number of cases / total cases | Unknown age Number of cases / total cases |
|---|---|---|---|
| Male Number of cases / total cases | 53/4309 (1.2%) | 1780/4309 (41.3%) | 202/4309 (4.7%) |
| Female Number of cases / total cases | 38/4309 (0.9%) | 40/4309 (0.9%) | 139/4309 (3.2%) |
| Non-binary Number of cases / total cases | 0/4309 (0.00%) | 1/4309 (0.0%) | 0/4309 (0.00%) |
| Unknown gender Number of cases / total cases | 87/4309 (2.0%) | 907/4309 (21.0%) | 1062/4309 (24.6%) |

## Number of incident cases by route of transmission

Most cases of mpox infection reasonably concluded by review authors to have resulted from a single transmission route occurred through direct physical sexual contact (Table 4). Where they could identify a single route of human-to-human transmission resulting in mpox infection, investigators reported no cases in which inhalation could reasonably have been the singular mode of mpox virus transmission (Table 4).

Nine studies reported cases of Clade IIa [31, 49,82,83,99,101,102,124,125] and four studies reported cases of Clade IIb [17,103,104,113]. Nineteen studies reported cases as West African clade in 2022 before the change in clade nomenclature in August 2022. Since they occurred in 2022, it is assumed these cases are likely to be clade IIb [13,14,22,29,47,51,57,63,78,79,86,91, 105,108,109,118,121,122,126]. Twenty-eight studies were published prior to 2022 that did not

**Table 2. Summary of findings: Respirator versus a medical mask for reducing mpox infection.**

**Population: Adults and children with mpox**
**Intervention: Respirator in addition to contact and droplet precautions**
**Comparator: Medical mask as part of contact and droplet precautions**
**Setting: Inpatient and outpatient**

| Outcome | Study results and measurements | Absolute effect estimates transmission | | Certainty of evidence | Comment |
|---|---|---|---|---|---|
| | | **Medical mask as part of contact and droplet precautions** | **Respirators in addition to contact and droplet precautions** | | |
| **Mpox infection** inferred from transmission route frequency data[a] | No reported cases of transmission by inhalation in 4309 patients (114 studies) Inferred odds ratio[b]: 1 | Uncertain[c] | Uncertain, but no different to medical mask[d] | ⊕⊕⊕⊖ Moderate[e] Due to indirectness | The use of a respirator probably has no difference in preventing mpox transmission compared to a medical mask. |

[a]No studies identified that directly informed the research question; data from route of transmission frequency resulting in mpox infection was inferred.

[b] Review findings indicated that mpox was transmitted in almost all occasions by direct physical contact; no cases of transmission through inhalation were identified. The review team inferred that if there are no or almost no cases of transmission by inhalation there would be no difference between the intervention and comparator groups.

[c]We could not estimate the baseline transmission risk due to absence of data.

[d]We could not estimate the risk in the intervention group due to an unknown baseline risk in the comparator group.

[e]Rated down one level for indirectness due to limited data on route of transmission frequency for Clade I mpox virus.

**Table 3. Summary of findings: Healthcare facility use of an airborne precaution room versus an adequately ventilated room for mpox patients for reducing mpox infection.**

**Population: Adults and children with mpox**
**Intervention: Airborne precaution room**
**Comparator: Adequately ventilated room**
**Setting: Inpatient and outpatient**

| Outcome | Study results and measurements | Absolute effect estimates transmission | | Certainty of evidence | Comment |
|---|---|---|---|---|---|
| | | Adequately ventilated single room | Airborne precaution room | | |
| Mpox infection inferred from transmission route frequency data[a] | No reported cases of transmission by inhalation in 4309 patients (114 studies) Inferred odds ratio[b]: 1 | Uncertain[c] | Uncertain, but no different to adequately ventilated room[d] | ⊕⊕⊕⊖ Moderate[e] Due to indirectness | The use of an airborne precaution room probably has no impact on preventing mpox transmission compared to an adequately ventilated room |

[a]No studies identified that directly informed the research question; data from route of transmission frequency resulting in mpox infection was inferred.

[b]Review findings indicated that mpox was transmitted in almost all occasions by direct physical contact; no cases of transmission through inhalation were identified. The review team inferred that if there are no or almost no cases of transmission by inhalation there would be no difference between the intervention and comparator groups.

[c]We could not estimate the baseline transmission risk due to absence of data.

[d]We could not estimate the risk in the intervention group due to an unknown baseline risk in the comparator group.

[e]Rated down one level for indirectness due to limited data on route of transmission frequency for Clade I mpox virus.

**Table 4. Incident cases of mpox by route of transmission.**

**Population: Adults and children with confirmed mpox**
**Setting: All settings**

| Route of transmission | Number of cases/ total cases[a] | Number of studies | References |
|---|---|---|---|
| Direct physical contact (sexual)[b] | 2366/4309 (54.9%) | 56 | 13,14,18,20,22,23,25–27,33,34,36,37, 39,40,42,44,45,47,48,51,58–61,63,65, 67,68,71,72,78,79,83,85–88,92,93,96,97, 103–107,109,110–112,114,115,119,126 |
| Direct physical contact (non-sexual) [b] | 6/4309 (0.1%)) | 2 | 35,121 |
| Fomite[b] | 2/4309 (0.0%) | 1 | 103 |
| Transplacental[b] | 1/4309 (0.0%) | 1 | 69 |
| Percutaneous injury with contaminated object[b] | 3/4309 (0.1%) | 3 | 28,30,70 |
| Inhalation[b] | 0/4309 (0.0%) | 0 | Not applicable |
| Multiple routes[c] | 1000/4309 (23.2%) | 30 | 16,17,19,21,24,31,32,35,41,43,49,50,53, 54,56,57,62,66,76,81,82,89,91,94,95, 102,105,117,120,123 |
| Unknown[d] | 931/4309 (21.6%) | 46 | 15,16,19,24,29,30,32,35,38,41,43,46,49, 52,55,62,65,69,73–75,77, 80,82–84,87,90, 93,98–102,105,108,110,113,116,118,120–125 |

Incident cases are defined as an individual changing from a state of non-disease to disease over a specific period of time reported within an included study.

[a]The denominator was calculated by the sum of all confirmed cases of human-to-human mpox transmission reported in included studies.

[b]A single route of transmission was identified as reasonably possible by review authors.

[c]More than one route of transmission was identified as possible by review authors. Possible transmission routes: direct physical sexual contact, direct physical non-sexual contact, fomite, inhalation. Insufficient information was reported in studies to assign or hypothesise any route of transmission by review authors.

report clade of included cases [15,16,19,21,24,35,38,46,50,53–56,62,66,69,73,74,76,77,80,81, 95,100,116,117,120,123]. Fifty-two studies were published in 2022 that did not report a clade of included cases [18,20,23,25–28,30,33,34,36,37,39–42,44,45,48,52,58–61,64,65,67,68,70– 72,75,84,85,87–90,92–94,96–98,106,107,110–112,114,115,119].

Most cases of mpox infection resulting from direct physical sexual contact as a single route of transmission were considered to be likely to be Clade IIb, the clade associated with the 2022–2023 mpox outbreak (Table 5). The two cases of mpox infection resulting from fomite transmission were Clade IIb. There were no cases of mpox virus Clade I reported in which a single route of transmission resulting in infection could reasonably be identified [32,43].

Reported cases by clade in which multiple routes of transmission were judged as reasonably possible by review authors, or the route of transmission was unknown, are available in the S4 Table.

Eight studies reported 120 cases of transmission resulting in mpox infection within a healthcare setting in which route of transmission may have been direct physical contact (non-sexual), fomite or inhalation [16,55,62,76,82,102,120,123]. Due to limited information reported by study authors, the review team were unable to disaggregate data further.

There were 538 cases that authors reported to have occurred within a household setting; however, due to limited reported information, no further disaggregation by route of transmission proved possible [21,16,19,24,32,49,50,52,54,56,62,76,81,120].Authors reported possible routes of human-to-human transmission resulting in infection as inhalation, fomite, direct sexual physical contact and direct physical non-sexual contact.

## Clinical and environmental sampling

Two studies [72,129] attempted viral isolation from respiratory tract samples of patients with mpox; mpox virus was isolated from saliva in 22/33 (66.7%) of samples between days 3 and 9 from symptom onset and in 1/4 (25%) oropharyngeal samples taken on day 9 from symptom onset (S5 Table).

Authors identified two studies in which viral isolation was attempted from air samples collected in hospital rooms solely occupied by individuals with mpox [128,129]; replication competent virus was identified in one air sample in one study [128] (S6 Table).

## Subgroups

**Health care worker transmission.**   Health care workers were the population of interest in review questions 1 and 2 (S1 and S2 Tables).

A healthcare worker in the United Kingdom in 2018 was diagnosed with mpox after changing the bed linen of a confirmed mpox patient using an apron and gloves; there was no direct physical contact with the mpox patient [116]. Investigators judged that transmission was possibly by fomite or inhalational route.

Three studies reported mpox infection in a healthcare worker through percutaneous injury with a contaminated sharp object that had been in contact with an mpox lesion [28,30,70].

**Patient-to-patient transmission.**   An outcome in review question 2 concerned the risk of transmission to patients (S2 Table). Two studies were identified that provided relevant data [53,62]. Jezek et al 1986 [53] reported mpox in a child in the Democratic Republic of the Congo who had visited a hospital several times where another child with confirmed mpox had been admitted. There was no known physical contact between the two. The child with subsequent mpox infection had walked past the mpox infected child in an outdoor space in the hospital grounds and past the mpox patient isolation area. The mpox infected child and the child who subsequently developed mpox had also received injections on the same day at the hospital

in which one syringe and 35 needles were being reused for all injections (study authors stated the two children had different needles). Review authors hypothesized fomite or inhalation transmission. Learned et al 2003 [62] reported a case of mpox in a patient hospitalized for malaria and in the same hospital as patients with mpox; authors reported no further information.

**Certainty of the evidence.** Certainty of evidence commenced at a rating of low due to inclusion of observational study designs [12]. Evidence from observational studies indicated that mpox was transmitted, in almost all occasions, by direct physical contact. We further found compelling evidence that mpox transmission by inhalation did not occur, or if it did, was extremely unusual. For review question 1, the logical inference was made that if there are no or almost no cases of transmission by inhalation, use of a respirator by a health care worker would prevent either none or very few mpox transmissions (Table 2). The evidence is therefore at least moderate certainty of little or no benefit in preventing transmission from respirator use. Similarly, for review question 2, the evidence is at least moderate that the use of an air-borne precaution room probably has little to no impact on preventing mpox transmission (Table 3). The certainty of the evidence for review questions 1 and 2 is thus based on logical inferences in this situation in which a formal comparison is lacking [12,135].

## Results: Review question 3

Review question 3 concerned IPC interventions related to the physical isolation of mpox patients with active lesions.

### Incident cases by route of transmission

In situations in which investigators could identify a single route of human-to-human transmission resulting in infection, 2366/4309 (54.9%) cases were transmitted via direct physical sexual contact, 6/4309 (0.1%) cases were transmitted via direct physical non-sexual contact, and 2/4309 (0.0%) cases via fomites (Table 2). Confirmed Clade IIb or cases likely to be Clade IIb form the majority of the data (Table 5).

The two cases of infection transmitted via fomites occurred in healthcare workers who visited a patient's home for one hour, wore personal protective equipment during the visit (N95 masks, eye protection, gowns), used gloves when taking clinical samples, and did not directly physically touch the patient. It was identified that some equipment used by the healthcare workers may not have been decontaminated before being handled without personal protective equipment [103].

### Clinical and environmental samples

Eight studies reported attempts to isolate virus from lesion samples. In four studies reporting the date of clinical sampling from symptom onset, 8/10 (80%) of lesion samples contained replication competent virus (S7 Table) [61,62,72,78]. In four studies in which the day of sampling was not documented, virus isolation was reported in 46.73% of lesion samples (S7 Table) [24,127,130,131].

Five studies attempted viral isolation from environmental surface samples [73,128,132–134]. Each study sampled high-touch surfaces (for example door handles and switches) and items that had been in close contact with infected persons (including towels and clothes). Studies conducted within hospitals included sampling of the anterooms in which personal protective equipment was doffed and disposed, and sampling of the personal protective equipment [128,133]. Gould et al 2022 [128] included sampling of a deposition area in each room which was unlikely to have been touched by patients or staff. Morgan et al 2022 [73] compared the

**Table 5. Proportion of incident cases of mpox by clade in which a single route of transmission was identified.**

Proportion of incident cases of mpox, by route of transmission and clade
Setting: All settings
Transmission: A single route of transmission was identified

| Route of transmission | Clade IIa number/ total cases[b] | Clade IIb number/ total cases[c] | Likely Clade IIb[a] number/ total cases[d] | Clade not reported in 2022 number/ total cases[e] | Clade not reported before 2022 number/ total cases[f] |
|---|---|---|---|---|---|
| Direct physical contact (sexual) | 12/12 (100.0%) 1 study [83] | 3/5 (60.0%) 2 studies [103,104] | 1083/1084 (99.9%) 12 studies [13,14,22,47,51,63,78,79,86,105, 109,126] | 1268/1271 (99.8%) 41 studies [18,20,23,25–27,33,34,36,37,39,40,42,44,45,48,58– 61,64,65,67,68,71,72,85,87,88,92,93,96,97,106,107,110– 112,114,115,119]) | 0/6 (0.0%) |
| Direct physical contact touch (non-sexual) | 0/12 (0.0%) | 0/5 (0.0%) | 1/1084 (0.1%) 1 study [121] | 0/1271 (0.0%) | 5/6 (83.3%) 1 study [35] |
| Fomite | 0/12 (0.0%) | 2/5 (40.0%) 1 study (103) | 0/1084 (0.0%) | 0/1271 (0.0%) | 0/6 (0.0%) |
| Transplacental | 0/12 (0.0%) | 0/5 (0.0%) | 0/1084 (0.0%) | 0/1271 (0.0%) | 1/6 (16.7%) |
| Inhalation | 0/12 (0.0%) | 0/5 (0.0%) | 0/1084 (0.0%) | 0/1271 (0.0%) | 0/6 (0.0%) |
| Percutaneous injury with contaminated object | 0/12 (0.0%) | 0/5 (0.0%) | 0/1084 (0.0%) | 3/1271 (0.2%) 3 studies [28,30,70] | 0/6 (0.0%) |
| Total cases (2378)[g] | 12 | 5 | 1084 | 1271 | 6 |

Incident cases are defined as an individual changing from a state of non-disease to disease over a specific period of time reported within an included study.

[a]Clade IIb is the primary variant largely circulating in the 2022 global mpox outbreak. These cases were reported in included studies in 2022 as West African clade before the change in clade nomenclature in August 2022. Since they occurred in 2022, it is assumed the cases are likely to be clade IIb.

[b]Denominator calculated as the sum of all reported Clade IIa mpox cases due to single route of transmission in included studies.

[c]Denominator calculated as the sum of all reported Clade IIb mpox cases due to single route of transmission in included studies.

[d]Denominator calculated as the sum of all mpox cases reported West African clade in 2022 due to a single route of transmission in included studies.

[e]Denominator calculated as the sum of all mpox cases due to single route of transmission in included studies without a clade reported in 2022.

[f]Denominator calculated as the sum of all mpox cases due to single route of transmission in included studies without a clade reported before 2022.

[g]Total 2378 incident cases of mpox in which a single route of transmission was identified.

frequency and quantity of virus detection from samples of non-porous and porous articles; authors reported that detection of viable virus was significantly more frequent from samples collected from porous materials. The frequency of detection of replication competent virus was between 0 and 60% in surface samples (S8 Table).

## Certainty of the evidence

Certainty of evidence commenced at a rating of low due to inclusion of observational study designs [12]. Evidence from observational studies indicated that mpox was transmitted, in almost all occasions, by direct physical contact. There were very few cases of fomite transmission. The logical inference was made that there may be minimal added benefits to physically isolating cases provided all lesions are covered, direct physical contact with others is avoided

**Table 6. Summary of findings: Physical case isolation until all lesions are fully healed versus no physical case isolation until all lesions are fully healed for reducing mpox infection.**

**Population: Adults and children with confirmed mpox**
**Intervention: Mpox patient does not physically isolate[a], covers all non-healed lesions, wears a medical mask**
**Comparator: Mpox patient physically isolated[a] until all lesions are fully healed**
**Setting: Household and community settings**

| Outcome | Study results and measurements | Absolute effect estimates transmission | | Certainty of evidence | Comment |
|---|---|---|---|---|---|
| | | Mpox patient isolated until all lesions are fully healed | Mpox patient does not isolate when all non-healed lesions are covered and wears a medical mask | | |
| **Mpox infection** inferred from transmission route frequency data[b] | 2366/4309 cases Direct physical sexual contact 6/4309 cases Direct physical non-sexual contact 2/4309 cases Fomite Inferred odds ratio[c]: 1 | Uncertain[d] | Uncertain[e] | Low to Moderate[f] Due to serious indirectness | Isolating patients probably does not prevent transmission of mpox compared to not isolating patients (provided all lesions are covered, a medical mask is worn and physical contact with others is avoided) |

[a]Physical isolation is defined as physical separation from other people.

[b]No studies identified that directly informed the research question; data from route of transmission frequency resulting in mpox infection was inferred.

[c] Review findings indicated that mpox was transmitted in almost all occasions by direct physical contact; there were very few cases of fomite transmission. The review team inferred that if all lesions were covered, direct physical contact with others is avoided, and a medical mask worn, there would be no difference between the intervention and comparator groups.

[d]We could not estimate the baseline transmission risk due to absence of data.

[e]We could not estimate the risk in the intervention group due to an unknown baseline risk in the comparator group

[f] Rated down one level for indirectness due to limited data on route of transmission frequency for Clade I mpox virus.

and a medical mask is worn (low to moderate certainty, downgraded one to two levels for serious risk of indirectness; Table 6). The certainty of the evidence for review question 3 is based on a logical inference in this situation in which a formal comparison is lacking [12,135].

## Discussion

We found no evidence from randomized controlled trials or observational comparative studies concerning airborne and droplet interventions or case physical isolation measures in mpox or mpox-like viruses capable of human-to-human transmission. Investigators reported no cases of mpox infection in which transmission by inhalation could reasonably be identified as the single route of transmission. Investigators reported 2 out of 4309 cases in which mpox virus could have reasonably been exclusively transmitted resulting in infection through fomites (103). In comparison, in 2366/4309 (54.9%) cases investigators identified transmission resulting in infection occurring through direct physical sexual contact (Table 4). Study investigators identified infectious mpox virus in saliva [129] and oropharyngeal swabs [72] and identified competent virus in 1/28 (3.6%) air samples [128,129]. Viral isolation was successful in 101/209 (48.3%) of lesion samples from 8 studies [24,61,62,72,78,127,130,131]; surface sampling in domestic and healthcare environments in 5 studies detected viable mpox virus in 16.2% of samples (range 0–60%) [73,128,132,133,134]. The presence of infectious virus in clinical samples and environmental samples provides only very low certainty evidence regarding risk of transmission that may lead to infection.

There is scarce data concerning transmission of mpox to health care workers. Mpox infection in healthcare workers was identified through percutaneous injury in three cases [28,30,70] and through possible fomite or inhalation in one case [116].

## Strengths and limitations

This review is strengthened by a comprehensive search strategy across multiple databases and authors independently assessed all studies for eligibility in duplicate to identify all possible relevant literature. Further, a clear conceptual data framework to address the research questions was undertaken; key data from all available literature that could inform the research questions was identified and synthesized in the two review stages.

This review is limited by the existing available evidence base on mpox. The review team utilized only broad search terms inclusive of terms for mpox-like viruses however no comparative interventional studies were identified. There is limited epidemiological evidence on the risk of fomite and inhalational transmission, and limited evidence on the infectious period for different routes of transmission. Inclusion of studies in English only may have influenced the completeness of findings. We expect publication bias to be sensitive to transmission routes resulting in infection; that is, if a route of transmission had been identified by investigators, this is likely to be published and captured for inclusion in the review and conversely, if a route of transmission had been not found, it is unlikely to be published. Another limitation of the review is the time elapsed since the literature search date (September 2022) and the call to topic experts in December 2022 to identify any further evidence. We are not aware of any systematic reviews addressing the same review questions covering the same scope or with as comprehensive inclusion criteria since the date of our literature search. To our knowledge there is also no known prior systematic review investigating airborne or droplet precautions or case home isolation IPC measures for mpox or mpox-like viruses. This review is an example of evidence synthesis methods in an area of scarce literature to answer key public health questions.

## Implications for practice and research

The findings of this review provide compelling evidence that transmission of mpox resulting in infection occurs primarily through direct physical contact. This finding agrees with a recent analysis of global surveillance data reporting the most common route of transmission in the 2022 mpox outbreak was direct physical sexual contact [1]. Secondary household attack rates are estimated to be 10% overall [136]. Household contact is the most common reported route of acquisition of infection amongst children, but sexual contact is the commonest reported route of transmission amongst adults [137]. Marshall et al [138] investigated exposures amongst 313 healthcare workers in different settings, noting duration and type of contact as well as personal protective equipment used. No cases of mpox resulted from a range of contacts including direct skin-to-skin contact with lesions and exposure to aerosol generating proceedures with or without FFP3/N95 masks. Most of these contacts were brief. It is difficult to distinguish between the transmission risk posed by close physical non-sexual contact and sexual contact. However more skin exposure, contact between mucous membranes or duration of contact may increase risk of transmission. In the 2022 global mpox outbreak, primary lesions commonly occurring at sites of sexual contact e.g. genital/anorectal or oral lesions and clustering of lesions at those sites support the conclusion that direct sexual contact is an important route of transmission [44]. In line with current guidance, avoidance of direct contact with skin lesions would likely reduce risk of transmission [131].

Epidemiological evidence and data from clinical and environmental sampling provides limited support for the hypothesis that inhalation or fomite modes of transmission are significant.

Replication competent virus has been identified in saliva; precautions to avoid direct exposure to respiratory droplets may be appropriate pending further data. The impact of airborne and droplet IPC measures in reducing transmission may be small (moderate certainty evidence). There is potential for shedding of infectious virus onto surfaces from lesions of detached scabs; covering mpox lesions is likely to reduce onward transmission however there is probably minimal reduction in transmission from added physical isolation of patients (moderate certainty evidence). Suitable cleaning protocols and caution around sharing items such as bedding or towels which may be contaminated is recommended in some settings.

Multiple factors such as route of exposure, infecting dose, susceptibility of the exposed individual would likely affect the relative risk of transmission resulting in infection. Currently, recommendations for airborne and droplet precautions and case home isolation IPC measures in mpox rely on expert opinion and inferences from data concerning transmission frequency by route of transmission [139].

## Conclusion

No available evidence from comparative interventional studies addressing airborne and droplet precautions and case home isolation IPC measures to prevent the transmission of mpox exists. Current findings suggest that transmission resulting in infection occurs primarily through direct physical contact. No cases of transmission resulting in infection via inhalation were identified; the impact of airborne and droplet IPC measures in reducing transmission may be minimal. Covering mpox lesions, wearing a medical mask and avoiding physical contact with others is likely to reduce onward transmission; there is probably minimal additional reduction in transmission from also physically isolating patients. Further research is needed into effective IPC measures to reduce the transmission of mpox, especially in the event of any future outbreak and in endemic settings.

## Supporting information

**S1 Checklist. PRISMA 2020 checklist.**
(DOCX)

**S1 Fig. Flow diagram of study selection.**
(TIF)

**S1 Table. Full details of review question 1.**
(DOCX)

**S2 Table. Full details of review question 2.**
(DOCX)

**S3 Table. Full details of review question 3.**
(DOCX)

**S4 Table. Incident cases of mpox by clade with multiple or unknown routes of transmission resulting in infection.**
(DOCX)

**S5 Table. Clinical samples of viral isolation attempts from adults or children with confirmed mpox infection.**
(DOCX)

**S6 Table. Air sampling in environments occupied by adults with confirmed mpox infection.**
(DOCX)

**S7 Table. Mpox lesion clinical samples in which viral isolation was attempted.**
(DOCX)

**S8 Table. Surface sampling in environments occupied by adults with confirmed mpox infection.**
(DOCX)

**S1 Appendix. Review search strategy.**
(DOCX)

**S2 Appendix. Quality appraisal questions.**
(DOCX)

**S3 Appendix. Quality appraisal results.**
(DOCX)

## Author Contributions

**Conceptualization:** Gordon Guyatt.

**Data curation:** Rebecca Kuehn, Tilly Fox, Vittoria Lutje, Susan Gould.

**Formal analysis:** Rebecca Kuehn, Tilly Fox, Susan Gould.

**Investigation:** Rebecca Kuehn.

**Methodology:** Rebecca Kuehn, Tilly Fox, Gordon Guyatt, Susan Gould.

**Project administration:** Rebecca Kuehn, Tilly Fox, Susan Gould.

**Resources:** Vittoria Lutje, Susan Gould.

**Software:** Vittoria Lutje.

**Supervision:** Gordon Guyatt.

**Writing – original draft:** Rebecca Kuehn, Tilly Fox, Gordon Guyatt, Susan Gould.

**Writing – review & editing:** Rebecca Kuehn, Tilly Fox, Gordon Guyatt, Vittoria Lutje, Susan Gould.

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
