## [Decision Letter · Decision Letter 0]

9 Jul 2023

PGPH-D-23-00439

Infection prevention and control measures to reduce the transmission of mpox: a systematic review

Dear Rebecca Kuehn,

Thank you for submitting your manuscript to PLOS Global Public Health. After careful consideration, we feel that it has merit but does not fully meet PLOS Global Public Health’s publication criteria as it currently stands. Therefore, we invite you to submit a revised version of the manuscript that addresses the points raised during the review process.

Reviewer's comments are attached

We look forward to receiving your revised manuscript.

Kind regards,

Sukanta Chowdhury, Ph.D

Academic Editor

Journal Requirements:

2. Please send a completed 'Competing Interests' statement, including any COIs declared by your co-authors. If you have no competing interests to declare, please state "The authors have declared that no competing interests exist".

3. Please provide a/amend your detailed Financial Disclosure statement. This is published with the article. It must therefore be completed in full sentences and contain the exact wording you wish to be published.

4. Please update the Funding Information in the system to reflect the details included in your updated Financial Disclosure Statement.

5. We do not publish any copyright or trademark symbols that usually accompany proprietary names, eg (R), (C), or TM  (e.g. next to drug or reagent names). Please remove all instances of trademark/copyright symbols throughout the text, including ® on Appendix.

6. We notice that your supplementary [figures/tables/Appendix] are included in the manuscript file. Please remove them and upload them with the file type 'Supporting Information'. Please ensure that each Supporting Information file has a legend listed in the manuscript after the references list.

Additional Editor Comments (if provided):

The manuscript is informative. I have added few more specific comments with reviewer that need to address.

Specific comments:

1. Abstract: Authors have mentioned the objective as “To assess the effectiveness of respiratory interventions…” As this is a review manuscript, authors have no scope to assess the effectiveness. Authors can use alternative words (e.g. review, summarize) instead of “assess”.

2. Abstract: Though the main objective was to assess the effectiveness of intervention but no information was included in result section about intervention.

3. Abstract: Transmission through inhalation is not reported as a single route. Why you suggested to wear medical mask to reduce transmission.

4. Introduction: No references are included. Why?

5. Methods: Please check the exclusion criteria. How many exclusion criteria did you consider? Any of the criteria?

6. Use reference for WHO Clinical Management and Infection Prevention and Control guideline

7. Provide a definition for “suspected case of mpox infection”.

8. Page 8: Data synthesis: “What is the proportion of new incident cases…...” Is it new cases or 2ndary cases? Why no cohort studies were considered for “What is the infectious period of mpox, disaggregated by route of transmission?”

9. Page 9: These following sentences are mentioned in method section.

“No randomized controlled trials or observational comparative studies were identified that directly informed any of the IPC review questions. No studies concerning mpox-like viruses met the inclusion criteria.” Avoid repeated information.

10. Page 10: the footnotes c and d are not properly identified

11. Page 11: the footnotes c and d are not properly identified

12. Table 3, 4 and 5: It is better to mention the study number.

13. Page 17: Inferred odds ratioc: 1 : Is it suggested by a study or is it your interpretation? If it its resulted from other study, please add reference.

14. Page 17: Which term is more appropriate? Incident cases or 2ndary cases? Human-human transmission is not always required for Incident cases.

15. Discussion: First paragraph: This is useful if you add some logics that support direct physical sexual contact increase the risk of mpox transmission.

16. Discussion: no discussion for suggested interventions to reduce mpox transmission. Please add a paragraph for that.

Reviewers' comments:

Reviewer's Responses to Questions

**Comments to the Author**

1. Does this manuscript meet PLOS Global Public Health’s publication criteria? Is the manuscript technically sound, and do the data support the conclusions? The manuscript must describe methodologically and ethically rigorous research with conclusions that are appropriately drawn based on the data presented.

Reviewer #1: Partly

2. Has the statistical analysis been performed appropriately and rigorously?

Reviewer #1: N/A

3. Have the authors made all data underlying the findings in their manuscript fully available (please refer to the Data Availability Statement at the start of the manuscript PDF file)?

Reviewer #1: Yes

4. Is the manuscript presented in an intelligible fashion and written in standard English?

Reviewer #1: Yes

5. Review Comments to the Author

Reviewer #1: Please find comments to the manuscript embedded in the attachment. The areas that are commented are highlighted.

For my response to Review Question 2, the authors could highlight in Discussion that uncertainties exist in the numeric results even though these uncertainties are hard to be quantified.

6. PLOS authors have the option to publish the peer review history of their article (what does this mean?). If published, this will include your full peer review and any attached files.

**Do you want your identity to be public for this peer review?** For information about this choice, including consent withdrawal, please see our Privacy Policy.

Reviewer #1: **Yes: **Dehao Chen

---

## [Decision Letter · Decision Letter 1]

13 Nov 2023

PGPH-D-23-00439R1

Infection prevention and control measures to reduce the transmission of mpox: a systematic review

Dear Dr. Rebecca Kuehn,

Thank you for submitting your manuscript to PLOS Global Public Health. After careful consideration, we feel that it has merit but does not fully meet PLOS Global Public Health’s publication criteria as it currently stands. Therefore, we invite you to submit a revised version of the manuscript that addresses the points raised during the review process.

We look forward to receiving your revised manuscript.

Kind regards,

Sukanta Chowdhury, Ph.D

Academic Editor

Journal Requirements:

1. Please update your online Competing Interests statement. If you have no competing interests to declare, please state: “The authors have declared that no competing interests exist.”

2. Please ensure that you refer to Figure 1 in your text as, if accepted, production will need this reference to link the reader to the figure.

3. Please include a separate legend for Figure 1 in your manuscript.

4. Please ensure that you cite or refer to Tables 2 and 3 in your text as, if accepted, production will need these references to link the reader to the tables.

5. We notice that your supplementary materials are included in the manuscript file. Please remove them and upload them with the file type 'Supporting Information'. Please ensure that each Supporting Information file has a legend listed in the manuscript after the references list.

6. Please add a full list of legends for all your Supporting Information files after the References list.

Additional Editor Comments (if provided):

Reviewers' comments:

Reviewer's Responses to Questions

**Comments to the Author**

1. If the authors have adequately addressed your comments raised in a previous round of review and you feel that this manuscript is now acceptable for publication, you may indicate that here to bypass the “Comments to the Author” section, enter your conflict of interest statement in the “Confidential to Editor” section, and submit your "Accept" recommendation.

Reviewer #1: (No Response)

2. Does this manuscript meet PLOS Global Public Health’s publication criteria? Is the manuscript technically sound, and do the data support the conclusions? The manuscript must describe methodologically and ethically rigorous research with conclusions that are appropriately drawn based on the data presented.

Reviewer #1: Partly

3. Has the statistical analysis been performed appropriately and rigorously?

Reviewer #1: (No Response)

4. Have the authors made all data underlying the findings in their manuscript fully available (please refer to the Data Availability Statement at the start of the manuscript PDF file)?

Reviewer #1: (No Response)

5. Is the manuscript presented in an intelligible fashion and written in standard English?

Reviewer #1: Yes

6. Review Comments to the Author

Reviewer #1: The authors have put significant effort in addressing comments raised from the initial round of the revision. Nonetheless, there are issues arise from the current revision that needs to be resolved before the manuscript being accepted. Please refer to the comments in the attached file and their corresponding highlighted sections.

7. PLOS authors have the option to publish the peer review history of their article (what does this mean?). If published, this will include your full peer review and any attached files.

**Do you want your identity to be public for this peer review?** For information about this choice, including consent withdrawal, please see our Privacy Policy.

Reviewer #1: No

---

## [Editor Report · Decision Letter 2]

29 Nov 2023

Infection prevention and control measures to reduce the transmission of mpox: a systematic review

PGPH-D-23-00439R2

Dear Rebecca Kuehn,

We are pleased to inform you that your manuscript 'Infection prevention and control measures to reduce the transmission of mpox: a systematic review' has been provisionally accepted for publication in PLOS Global Public Health.

Best regards,

Sukanta Chowdhury, Ph.D

Academic Editor